# Mechanistic Signatures of Human Papillomavirus Insertions in Anal Squamous Cell Carcinomas

**DOI:** 10.3390/cancers11121846

**Published:** 2019-11-22

**Authors:** Adeline Morel, Cindy Neuzillet, Maxime Wack, Sonia Lameiras, Sophie Vacher, Marc Deloger, Nicolas Servant, David Veyer, Hélène Péré, Odette Mariani, Sylvain Baulande, Roman Rouzier, Maud Kamal, Elsy El Alam, Emmanuelle Jeannot, Alain Nicolas, Ivan Bièche, Wulfran Cacheux

**Affiliations:** 1Institut Curie, Pharmacogenomic Unit, 26 rue d’Ulm, 75248 Paris, France; a.morel@baclesse.unicancer.fr (A.M.); sophie.vacher@curie.fr (S.V.); ivan.bieche@curie.fr (I.B.); 2Institut Curie, Medical Oncology Department, Versailles Saint-Quentin University, 35 rue Dailly, 92210 Saint-Cloud, France; roman.rouzier@curie.fr (R.R.); wulfran.cacheux@curie.fr (W.C.); 3Département d’Informatique Médicale, Biostatistiques et Santé Publique, Hôpital Européen Georges Pompidou, and Assistance Publique-Hôpitaux de Paris, 75015 Paris, France; maxime.wack@aphp.fr; 4Centre de Recherche des Cordeliers, INSERM UMRS1138, Université Paris Descartes, Sorbonne Paris Cité, 75006 Paris, France; 5Institut Curie, Genomics of Excellence (ICGex) Platform, PSL Research University, 26 rue d’Ulm, 75248 Paris CEDEX 05, France; Sonia.Lameiras@curie.fr (S.L.); sylvain.baulande@curie.fr (S.B.); 6Institut Curie, Bioinformatics and Computational Systems Biology of Cancer, PSL Research University, Mines Paris Tech, INSERM U900, 75248 Paris, France; marc.deloger@curie.fr (M.D.); nicolas.servant@curie.fr (N.S.); 7Laboratoire de virologie, Hôpital Européen Georges Pompidou, and Assistance Publique-Hôpitaux de Paris, 75015 Paris, France; david.veyer@aphp.fr (D.V.); helene.pere@aphp.fr (H.P.); 8Institut Curie, Centre de Ressources Biologiques, 26 rue d’Ulm, 75248 Paris, France; odette.mariani@curie.fr; 9Institut Curie, Department of Drug Development and Innovation (D3i), Institut Curie Paris & Saint Cloud, 75248 Paris, France; maud.kamal@curie.fr; 10Institut Curie, Pathology Department, 35 rue Dailly, 92210 Saint-Cloud, France; elsy.elalam@curie.fr; 11Institut Curie, Pathology Department, 26 rue d’Ulm, 75248 Paris, France; emmanuelle.jeannot@curie.fr; 12Institut Curie, PSL Research University, CNRS UMR3244, 75248 Paris, France; alain.nicolas@curie.fr; 13INSERM U1016, Université Paris Descartes University, 75006 Paris, France; 14Hôpital Privé Pays de Savoie, Service d’oncologie Médicale, 19 Avenue Pierre Mendès France, 74100 Annemasse, France

**Keywords:** anal cancer, HPV, integration, *NFIX*, prognostic factor

## Abstract

The role of human papillomavirus (HPV) in anal squamous cell carcinoma (ASCC) carcinogenesis has been clearly established, involving the expression of viral oncoproteins and optional viral DNA integration into the host genome. In this article, we describe the various mechanisms and sites of HPV DNA insertion and assess their prognostic and predictive value in a large series of patients with HPV-positive ASCC with long-term follow-up. We retrospectively analyzed 96 tumor samples from 93 HPV-positive ASCC patients using the Capture-HPV method followed by Next-Generation Sequencing, allowing determination of HPV genotype and identification of the mechanisms and sites of viral genome integration. We identified five different mechanistic signatures of HPV insertions. The distribution of HPV signatures differed from that previously described in HPV-positive cervical carcinoma (*p* < 0.001). In ASCC samples, the HPV genome more frequently remained in episomal form (45.2%). The most common signature of HPV insertion was MJ-SC (26.9%), i.e., HPV–chromosomal junctions scattered at different loci. Functionally, HPV integration signatures were not associated with survival or response to treatment, but were associated with viral load (*p* = 0.022) and *PIK3CA* mutation (*p* = 0.0069). High viral load was associated with longer survival in both univariate (*p* = 0.044) and multivariate (*p* = 0.011) analyses. Finally, HPV integration occurred on most human chromosomes, but intragenic integration into the *NFIX* gene was recurrently observed (*n* = 4/51 tumors). Overall, the distribution of mechanistic signatures of HPV insertions in ASCC was different from that observed in cervical carcinoma and was associated with viral load and *PIK3CA* mutation. We confirmed recurrent targeting of *NFIX* by HPV integration, suggesting a role for this gene in ASCC carcinogenesis.

## 1. Introduction

Anal squamous cell carcinoma (ASCC) accounts for about 2.5% of all gastrointestinal malignancies [1]. It is more frequent in women and in older patients (age ≥ 65). Although considered to be a rare tumor, its incidence is increasing, particularly in men [2,3]. ASCC is associated with human papillomavirus (HPV) infection. High-risk HPV (mainly, genotype 16) DNA is detected in up to 90%–95% of ASCC specimens [4,5,6]. Anal intercourse and a high lifetime number of sexual partners increase the risk of persistent HPV infection and subsequent anal intraepithelial neoplasia and ASCC [7]. Other risk factors for ASCC include immunosuppression induced by human immunodeficiency virus (HIV) infection or the use of immunosuppressants after solid organ transplantation, hematologic malignancies, as well as a history of other HPV-related cancers, autoimmune disorders, low socio-economic status, and smoking [7].

Radiotherapy (RT) alone or in combination with chemotherapy (chemo-radiotherapy, CRT) is the standard first-line treatment for nonmetastatic ASCC [7,8]. Local failure or recurrence (30%) is associated with poor prognosis [7]. Very few treatment options with only limited efficacy are available and are mostly based on salvage surgery with permanent colostomy and chemotherapy [7,8,9]. HPV-positive tumors display higher sensitivity to RT and are associated with better prognosis [10,11,12]. There is an urgent medical need to improve the prognosis and therapeutic stratification of ASCC patients by the identification of new prognostic factors, predictive biomarkers for response to RT/CRT, and therapeutic targets.

The role of HPV in tumor initiation and progression of cervical, anal canal, oropharynx, vulva, and penis carcinomas has been clearly established [13]. HPV is a small double-strand DNA virus, of approximately 8000 base pairs, with more than 150 different genotypes. The viral genome is either maintained as extrachromosomal multicopy nuclear episomes or integrated into the host genome [14,15]. Viral oncogenesis involves the expression of E6 and E7 proteins, which are able to interact with and inhibit p53 and pRb tumor suppressor proteins [16]. Integration of HPV DNA into the host genome has also been reported as a potential pathogenic mechanism [17].

We have recently developed a sensitive HPV double capture method (Capt-HPV) followed by Next-Generation Sequencing (NGS) in order to precisely and robustly determine HPV genotype, status (episomal versus integrated), viral load as well as the integration sites in the human genome [18]. This comprehensive approach was validated in a series of 72 cases of cervical squamous cell carcinoma using both tumor tissue biopsy and circulating viral DNA from blood samples [18] and identified five mechanistic signatures of HPV status: episomal [EPI]; integrated in a truncated form revealing two HPV-chromosomal junctions (2J) either colinear [2J-COL] or noncolinear [2J-NL], or multiple (>2) hybrid junctions [MJ] clustering in a single chromosomal region [MJ-CL] or scattered over different chromosomal regions [MJ-SC] of the human genome.

The aim of this study was to describe and characterize the HPV integration signatures using our Capt-HPV method in a large, single-institution, clinically annotated cohort of 93 patients with HPV-positive ASCC. We report the mechanistic HPV insertion signatures, assess their prognostic and predictive value and compare the HPV features of anal vs. cervical cancers.

## 2. Materials and Methods

### 2.1. Patients

Ninety-six cryopreserved HPV-positive tumor biopsy specimens from 93 patients treated for ASCC at the Institut Curie Hospital between 1993 and 2017 were retrospectively retrieved from the institutional tissue bank. Matched initial diagnostic biopsy and recurrence biopsy samples were available for three patients.

In accordance with French regulations and ethical requirements, patients were informed about the research performed on tissue specimens and did not express their opposition. The study protocol was approved by French Ethical Committee (Agreement number D-750602, France) and the ethics committee of the Institut Curie (Agreement number C75-05-18). Clinical data were collected retrospectively from electronic patient files.

### 2.2. Methods

#### 2.2.1. DNA Extraction

Tumor DNA was extracted from frozen tumor samples using previously published procedures [19]. Samples had full face sectioning performed in with Tissue-Tek optimal cutting temperature (O.C.T) compound to estimate the percentage of malignant epithelial nuclei in the sample relative to stromal nuclei. Macrodissection was performed if required to excise areas of nonmalignant tissue. Total genomic DNA was extracted with phenol-chloroform after proteinase K digestion, followed by the precipitation of nucleic acids in ethanol. DNA was quantified using Nanodrop spectrophotometer ND-1000 (ThermoScientific, Wilmington, DE, USA) and Qubit BR DNA assay (Invitrogen, Carlsbad, CA, USA).

#### 2.2.2. HPV Genotyping

HPV detection and genotyping were first performed using Real-Time PCR as previously described [6]. From 1998 to 2013, all samples were analysed by PCR using specific primers to identify HPV16, 18, 33, 45, 6, and 11 types and using GP5+/GP6+ primers to detect HPV L1 DNA [20]. After 2013, real-time PCR using Sybr Green (Roche Diagnostics, Mannheim, Germany) and specific primers for HPV16, 18, and 33 and the human GAPDH gene was performed on a 7900HT Fast Real-Time PCR System (Applied Biosystems). HPV L1 amplicons from HPV16-, 18-, and 33-negative samples were sequenced by Sanger method with GP6+ primer and HPV type identification was performed by alignment of the sequence with HPV sequence references, using the nucleotide blast program from NCBI (http://blast.ncbi.nlm.nih.gov/Blast.cgi).

#### 2.2.3. DNA Library Preparation

The DNA libraries were prepared using 500 ng of genomic DNA, starting with ultrasonication (Covaris, Woburn, MA, USA) to produce double-strand DNA fragments with an average length of 280 bp. End-Repair and A-tailing were then applied to facilitate ligation of the adapters, containing unique barcodes for each sample, specific to the Illumina technology for amplification and sequencing. KAPA Hyper Prep kit was used for these steps, according to the manufacturer’s instructions.

#### 2.2.4. HPV Double Capture Method

Capt-HPV was performed as previously described [18]. Briefly, the double capture method was carried out using the SeqCap EZ Rapid Library Small Target Capture method, developed by Roche NimbleGen, which is adapted to capture small DNA targets. The DNA libraries were multiplexed (by 12) and hybridized for 16 h with biotinylated HPV oligonucleotide probes, recognizing all HPV genotypes. DNA sequences were then captured by streptavidin beads and amplified by PCR. Double capture (i.e., two rounds of hybridization and capture) was performed to improve the efficiency and specificity. Postcapture libraries were sequenced using Illumina MiSeq system, with 150 paired-end reads and 24 samples multiplexed on a V2 micro flow-cell.

#### 2.2.5. HPV Viral Load

Viral load was estimated by the ratio of the number of reads mapped to HPV over the number of reads mapped to the human *KLK3* gene used as a control. All probes were included in the capture set. The optimal cut-off was defined as 12 based on the ROC (receiver operating characteristic) curve (see Statistical analysis).

#### 2.2.6. Bioinformatics Pipeline for Analysis of HPV Genotypes and Viral–Cellular Junctions

We used an automated alignment tool (Wack et al., manuscript in preparation) allowing HPV genotyping and viral–human DNA junction identification using specific filters. The insertion site was analyzed using reads containing both HPV and non-HPV sequences. The non-HPV sequence was aligned on GRCh37-hg19 (BLAT tool).

### 2.3. Statistical Analysis

Correlations between HPV integration signatures and clinical, laboratory and molecular features were analyzed using Chi-square tests, Chi-square tests with Yates’ correction or Fisher’s exact tests, as appropriate. Given the small sample size, 2J-COL and 2J-NL simple junctions and MJ-CL and MJ-SC multiple junctions were pooled for analyses. To visualize the efficacy of a molecular marker (i.e., HPV viral load) to discriminate two populations (patients who died/did not die) in the absence of an arbitrary cut-off value, data were summarized in an ROC curve [21]. The AUC (area under curve) was calculated as a single measure for discriminate efficacy. Overall survival (OS) was defined as the time interval from the date of ASCC diagnosis to death. Survival data were censored at the date of last follow-up. Patients with short follow-up (≤30 days, *n* = 3) were excluded from survival analyses. Survival curves were estimated by the Kaplan–Meier method, and compared using the Log-Rank test. The Cox proportional hazards regression model was used to assess prognostic significance and the results are presented as hazard ratios and 95% confidence intervals (CIs). For all statistical tests, the limit of significance was defined as *p* < 0.05.

## 3. Results

### 3.1. Patient Characteristics

Ninety-three HPV-positive ASCC patients were included. Patient characteristics are presented in Table 1. Briefly, 75 (80.6%) patients were women and eight (8.8%) patients had concomitant HIV infection. Fifteen (16.1%) patients were treated by upfront surgery, 63 (67.7%) patients were treated by CRT, and 22 (23.7%) received RT alone or in addition to surgery. Seventy-two (86.7%) of the 83 patients treated by RT or CRT achieved a complete tumor response (CR). Tumor biopsy was obtained prior to treatment in 54 (58.1%) patients and from recurrent disease after initial RT or CRT in 39 (41.9%) patients. Tumor samples at diagnosis and at disease recurrence were both available for three patients; a total of 96 samples were therefore analyzed.

### 3.2. HPV Genotype and Mutational Analysis

HPV genotype determined by PCR was confirmed by NGS, as seen in Table 1. HPV16 was the most common genotype (85/93, 91.4%), as previously reported [4,5,6]. The other HPV genotypes (8/93, 8.6%) were HPV6 (*n* = 3), HPV18 (*n* = 2), HPV33 (*n* = 1), HPV35 (*n* = 1), and HPV67 (*n* = 1). Previous mutational analysis of the *PIK3CA* gene performed on all tumor samples demonstrated that 22 patients (23.7%) harbored a *PIK3CA* hot spot mutation in exon 9 (*n* = 21) or exon 20 (*n* = 1) [6].

### 3.3. HPV Viral Load

Patients were classified as low (ratio < 12, *n* = 41) vs. high viral load (ratio ≥ 12, *n* = 52), determined as described in “Statistical analysis” section. No association was observed between HPV viral load and clinicopathological features, as seen in Appendix A.

### 3.4. Integration Mechanisms

Bioinformatics analysis of Capt-HPV reads allowed complete identification of HPV genome status. In the absence of integration (42/93, 45.2%), no HPV breakpoint or HPV–chromosomal junction reads were identified, indicating that the viral genome was maintained in its episomal form (EPI). In the presence of integration, the viral genome and its breakpoints as well as the coordinates of the integration sites in the human genome were identified by pure HPV reads and junction reads. All patients displaying HPV integration (51/93, 54.8%) exhibited partial and variable integration of the viral genome, albeit systematically retaining the HPV E6 and E7 oncogenes. The mechanistic integration signatures were distributed as follows: MJ-SC (*n* = 25, 26.9%), MJ-CL (*n* = 9, 9.7%), 2J-COL (*n* = 7, 7.5%), 2J-NL (*n* = 2, 2.2%), and a few other cases (*n* = 8, 8.6%) in which a single HPV-chromosomal junction was observed. The mechanistic distribution of HPV signatures was therefore markedly different from that observed in cervical carcinoma (*p* = 0.0000012), as seen in Figure 1.

### 3.5. Dynamic Evolution of Integration Status in Patients with Matched Samples

Among the three patients with matched pretreatment and recurrence biopsy samples, two had initial EPI status and remained EPI at disease recurrence, and one (33.3%) switched from EPI to 2J insertion signature, indicating that HPV integration status may change over time.

### 3.6. Insertion Sites

Multiple integration regions in the human genome were observed. We identified more than 90 HPV–chromosomal junction regions (intergenic or intragenic) in the 51 tumors with HPV integration, as seen in Figure 2. At the nucleotide level, all integration regions were unique and distributed throughout the chromosomes, except for small chromosomes 21, as seen in Figure 2. No insertion hot-spot was detected except for the *NFIX* gene (19p13.2) that was targeted in four tumors (two in intron 1 and two in intron 2), as seen in Figure 2 and Appendix A. In addition, we also found three other integration events adjacent to the *NFIX* gene but nonintragenic, of unknown significance, as seen in Figure 2.

### 3.7. Association between Insertion Mechanisms and Patient Clinicopathological Data

The distribution of HPV status and integration signatures (grouped into EPI, 2J and MJ cases) according to the patients’ clinical, laboratory, and pathological characteristics is presented in Table 2 and Appendix A. No significant association was observed between HPV integration signatures and clinicopathological features. In particular, no difference was observed between treatment-naive and recurrent disease groups (*p* = 0.93), as well as according to treatment type (RT or CRT, *p* = 0.76), and in HIV-positive vs. negative groups (*p* = 0.92), as seen in Table 2 and Appendix A.

Of note, samples with 2J insertion displayed low viral load, while MJ insertion was associated with high viral load (*p* = 0.022). Moreover, samples with viral integration (2J or MJ) more frequently harbored *PIK3CA* activating mutations (7/15, 46.7% and 11/36, 30.6%, respectively) than those harboring episomal HPV (4/42, 9.5%) (*p* = 0.0069).

### 3.8. Response to Treatment and Overall Survival

No significant association was observed between HPV integration signatures and response to RT/CRT (CR vs. non-CR, overall *p* = 0.36), as seen in Table 2. No correlation was found in subgroup analyses in the RT (*p* = 0.53) and CRT subgroups (*p* = 0.47), and in the HIV-negative subgroup (*p* = 0.70).

Ninety patients were evaluable for survival (follow-up > 30 days). Median follow-up was 46.2 months (range: 7.0–277.0 months). Tumor differentiation (*p* = 0.014), HPV16 genotype (*p* = 0.005), and CR after RT/CRT (*p* < 0.001) were significantly associated with OS in univariate analysis; in multivariate analysis, HPV16 genotype (*p* = 0.008), tumor stage (*p* = 0.048), and complete response after RT/CRT (*p* = 0.001) were independent prognostic factors, as seen in Table 1. A low viral load was associated with shorter OS both in univariate (HR: 1.89, *p* = 0.044, Figure 3) and multivariate (*p* = 0.011) analyses, as seen in Appendix A, but no correlation was observed between HPV integration signatures (EPI, 2J, and MJ) and OS (*p* = 0.50; Appendix A). The same findings were observed after exclusion of HIV-positive patients (HIV-negative subgroup: HR: 2.04, *p* = 0.038 for viral load, Appendix A, *p* = 0.70 for integration signature). The association between viral load and OS was stronger in the RT subgroup (HR: 6.30, *p* = 0.0006 for viral load, Appendix A, *p* = 0.53 for integration signature) vs. the CRT subgroup (nonsignificant trend, HR: 1.29, *p* = 0.54 for viral load, Appendix A, *p* = 0.49 for integration signature), and in the CR subgroup (HR: 3.09, *p* = 0.0022 for viral load, Appendix A, *p* = 0.69 for integration signature) vs. the non-CR subgroup (nonsignificant trend, HR: 2.38, *p* = 0.25 for viral load, Appendix A, *p* = 0.30 for integration signature).

Remarkably, four patients harbored HPV integration in the *NFIX* gene (4/51, 7.8%). These patients had more advanced tumor stage (*p* = 0.00016, with less stage I-II) and more frequently displayed MJ-SC or MJ-CL integration signatures (*p* = 0.036), but no correlation was observed with viral load (*p* = 0.63), as seen in Appendix A. In addition, all of these patients displayed complete response after RT/CRT and no metastasis, as seen in Appendix A. They tended to have longer OS compared to other patients (*p* = 0.093 and *p* = 0.051 in the whole cohort and in the subgroup of 50 patients with HPV integration evaluable for survival, respectively; Figure 4).

## 4. Discussion

In this study, we analyzed a large clinically annotated cohort of 93 HPV-positive ASCC patients for their HPV status and integration signatures using our Capt-HPV double capture method followed by NGS [18]. This HPV-unbiased and highly sensitive method allows (i) confirmation of HPV subtype, predominantly HPV16 in ASCC, determined by PCR, (ii) characterization of the tumor HPV sequence which is full-length when HPV remained episomal or variably truncated when HPV was integrated, (iii) quantification of HPV viral load, and (iv) analysis of the sites and mechanisms of HPV insertion into the human genome. Capt-HPV data retrieved from ASCC and cervical squamous cell carcinoma biopsies were consistent, but with a different distribution of insertion signatures [18]. Our analysis did not reveal any significant correlation between HPV integration signatures and known prognostic factors, except for *PIK3CA* mutation (*p* = 0.0069) and viral load (*p* = 0.022). HPV integration signatures also had no impact on OS or response to RT/CRT.

Our analysis of HPV integration signatures in ASCC revealed the predominance of EPI (42/93, 45.2%) over MJ (MJ-SC: 25/93, 26.9%, MJ-CL: 9/93, 9.7%) and 2J (2J-COL: 7/93, 7.5%, 2J-NL: 2/93, 2.2%) insertions. This distribution was statistically different (*p* < 0.001) from that observed in cervical squamous cell carcinoma, with a higher proportion of EPI in ASCC (45.2% vs. 29% in cervical squamous cell carcinoma) [18]. In oropharyngeal squamous cell carcinoma, Olthof et al. [22] reported that HPV was integrated in the host genome in only 39% of cases and remained in the episomal form in 61% of cases. In another study in head and neck squamous cell carcinoma (HNSCC, not limited to oropharyngeal tumor site), 35 out of 279 tumors displayed evidence of high-risk HPV types 16, 33, or 35 [23]. Twenty-five cases (71%) had integration of the viral genome into one or more locations in the human genome. Viral integration was associated with increases in somatic DNA copy number and a nonsignificant trend toward a higher mutational burden, suggesting that tumors with HPV integration might be associated with more immunogenic tumors [23]. The overall mechanistic HPV pattern therefore appears to vary according to the primary tumor tissue. In addition, one patient with matched pretreatment and recurrent disease biopsies switched from EPI to 2J insertion signature, suggesting that HPV integration status may change over time and in response to treatment.

Of note, samples with 2J insertions were associated with low viral load, while MJ samples were associated with high viral load (*p* = 0.022), suggesting that viral load is linked to the alternative mechanistic mode of HPV insertion proposed by Holmes et al. [18]. Moreover, although HPV integration signatures had no prognostic impact in our cohort, low viral load was an independent prognostic factor predictive of shorter OS in our cohort (*p* = 0.011). Similar findings have been reported in cervical cancer, where low HPV viral load is associated with shorter survival and poor prognosis features including with poorly differentiated tumors and metastatic lymph node metastasis [24,25,26]. Kim et al. [25] proposed increased E6/E7 oncoproteins as an underlying mechanism of lower viral replication and poor radiotherapy outcome in patients with low viral load: when HPV integrates into the host genome, this event disrupts the E2 open reading frame and results in increased expression of the E6 and E7 oncoproteins (which have been shown to contribute to enhanced therapy resistance [27]), while the actual viral DNA copy number is reduced as the production of infectious virus is aborted. Similarly, in ASCC, full integration of the HPV genome (estimated by the E2/E6 ratio) was associated with a low viral load compared with cases containing either episomes or mixed integrated and episomal DNA [28]. Therefore, HPV integration is hypothesized to modulate viral load and resistance to RT, and tumors with high tumor load may display favorable biological behavior and increased sensitivity to RT/CRT, even though we did not observe such correlation in our study.

We also observed that HPV16 is an independent predictor of survival in our patient cohort. HPV-16 is the most frequent genotype found in ASCC, and HPV-positivity is associated with longer survival and response to RT/CRT, similar to HNSCC [4,5,6,29]. In the study by Kim et al. [25] in cervical cancer, non-HPV-16 genotypes (particularly, HPV-18) have been associated with poor prognosis and low viral load, but the effect on RT outcome could not be assessed due to small sample size of the non-HPV-16 subgroup. We faced the same limitation in our study.

Interestingly, samples with viral integration (2J and MJ) more frequently harbored *PIK3CA*-activating mutations than those harboring episomal HPV (*p* = 0.0069). It has been suggested that HPV creates a selective pressure that promotes the emergence of tumors with APOBEC-mediated driver mutations in HNSCC [30]. The APOBEC (Apolipoprotein B mRNA Editing Catalytic Polypeptide-like) proteins are a family of evolutionarily conserved cytidine deaminases, involved in mRNA editing. These enzymes, when dysregulated, are a major source of mutations. In particular, APOBEC activity is responsible for the generation of *PIK3CA* gene mutation across multiple cancers, which may explain the positive association between HPV integration and *PIK3CA* mutation in our series [30]. In addition, in HPV-positive HNSCC, APOBEC editing signatures were strongly linked to overall mutational burden, suggesting that these tumors may be more immunogenic and candidates for immune therapy [31].

We observed that HPV insertion occurred in both intergenic regions and in repeated sequences, as well as in intragenic regions. In HPV-positive cervical cancer, viral genome integrations occur on all chromosomes, with no detectable hot spots, except for the *KLF5/KLF12* gene and adjacent to the *MYC* gene locus [18]. These recurrent insertions were not detected in HPV-positive ASCC. In contrast, integration into the *NFIX* gene was observed in four cases, confirming the results from the recent study by Jeannot et al. [15], along with three other integration events adjacent to the *NFIX* gene but nonintragenic, of unknown significance. *NFIX* belongs to the *NFI* gene family, consisting of four transcription factors (*NFIA*, *NFIB*, *NFIC*, and *NFIX*), involved in the regulation of cell proliferation and differentiation. The C-termini of the *NFI* protein family diverge due to the existence of multiple splicing sites. Each *NFI* transcription factor encodes multiple splicing variants, the function of which remains unknown in many cases. *NFI* transcription factors have both oncogenic and tumor suppressor potential, depending on the primary tumor site [32]. In lung cancer, silencing of *NFIX* induced a decrease in *IL6ST*, *TIMP1*, and *ITGB1* gene expression and reduced cell proliferation, migration, and invasion processes [33]. In colorectal cancer, high NFIX expression levels were associated with chemotherapy resistance [34]. In contrast, in esophageal squamous cell carcinoma, low NFIX expression was associated with positive lymph node metastasis and advanced tumor-node-metastasis stage [35]. In our study, HPV integration in the *NFIX* gene was associated with a trend towards longer OS in patients with HPV-positive ASCC (*p* = 0.05). We could not assess the effect of HPV integration on NFIX expression and function. The favorable survival trend that we observed may be due to the fact that these all of these patients achieved complete response after RT/CRT and none of them developed metastasis, both of which are favorable prognostic factors. This result, obtained on a small number of patients (4/51), needs to be validated in larger series, and further exploration of the underlying molecular and biological mechanisms is warranted.

## 5. Conclusions

In conclusion, the present study validates the robustness of our method of double capture followed by NGS for comprehensive one-step analysis of tumor HPV status. The distribution of the various HPV integration signatures differed between ASCC and cervical cancers and was associated with viral load and *PIK3CA* mutation [18]. The small number of events in this single-center study may have limited the statistical power to determine whether HPV status and integration signatures have a prognostic or predictive impact in HPV-positive ASCC, but was nevertheless sufficient to confirm recurrent targeting of *NFIX* by HPV insertion in ASCC, a potential favorable prognostic factor [15]. Further studies on larger, multicenter cohorts of anal and other HPV-associated tumors are warranted.

## Figures and Tables

**Figure 1 cancers-11-01846-f001:**
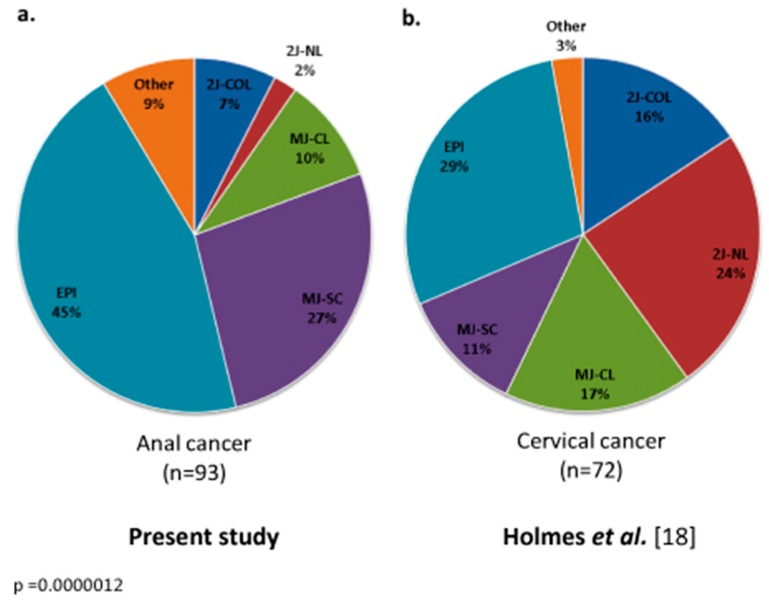
Distribution of HPV integration signatures according to the primary tumor site of HPV-positive squamous cell carcinoma: (**a**) anal cancer; (**b**) cervical cancer [18]. Chi-square test, *p* value for global comparison of EPI group vs. MJ-SC group vs. MJ-CL group vs. 2J-NL group vs. 2J-COL group vs. Other group between anal cancer and cervical cancer, *p* = 0.0000012. 2J-COL: two hybrid colinear junctions; 2J-NL: two hybrid nonlinear junctions; EPI: episomal; MJ-CL: multiple hybrid junctions clustered in one locus; MJ-SC: multiple hybrid junctions scattered at distinct loci.

**Figure 2 cancers-11-01846-f002:**
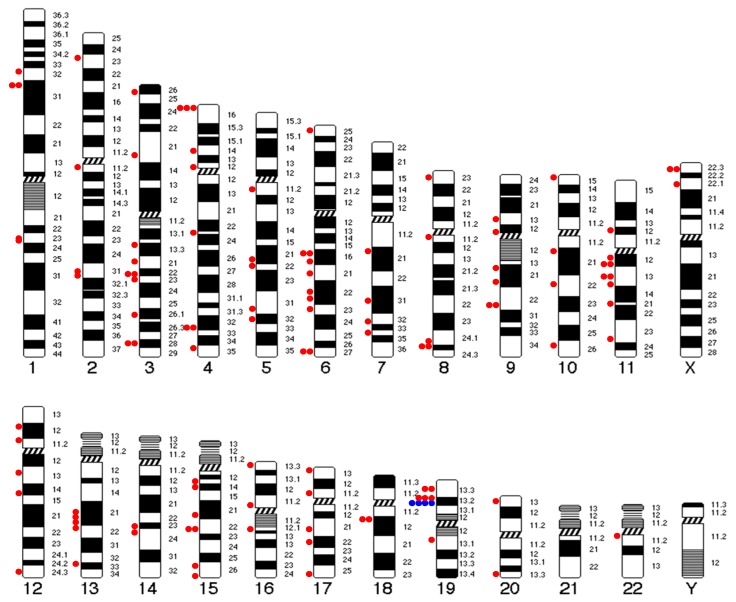
Distribution of HPV insertion sites in the genome of patients with HPV-positive anal squamous cell carcinoma (ASCC). Each dot represents an HPV integration site. Four intragenic integration events in the *NFIX* gene located at 19p13.2 were identified (blue dots) as well as three other integration events adjacent to the *NFIX* gene but nonintragenic (red dots), of unknown significance.

**Figure 3 cancers-11-01846-f003:**
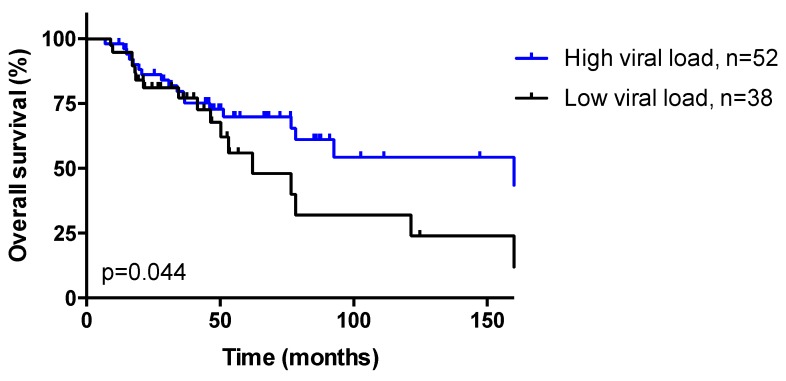
Overall survival (OS) curves of the 90 patients with HPV-positive anal squamous cell carcinoma (ASCC) evaluable for survival, according to HPV viral load. Median OS: 62.2 months in low (<12) viral load vs. 168.2 months in high (≥12) viral load. Log-Rank test, hazard ratio (HR): 1.89, *p* = 0.044.

**Figure 4 cancers-11-01846-f004:**
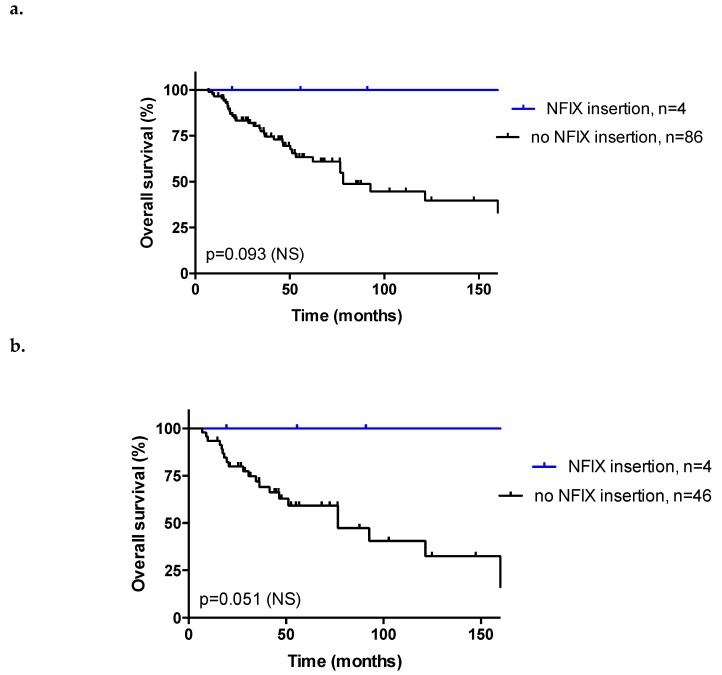
Overall survival (OS) curves in patients with anal squamous cell carcinoma (ASCC) and HPV insertion into *NFIX* gene; (**a**) in the overall cohort (*n* = 90 evaluable patients). Median OS: not reached in patients with *NFIX* insertion vs. 78.3 months in patients without *NFIX* insertion. Log-Rank test, *p* = 0.093. (**b**) in the subgroup of patients with HPV integration (*n* = 50 evaluable patients). Median OS: not reached in patients with *NFIX* insertion vs. 76.7 months in patients without *NFIX* insertion. Log-Rank test, *p* = 0.051. NS: not significant.

**Table 1 cancers-11-01846-t001:** Clinical and biological characteristics of the 93 patients with HPV-positive anal squamous cell carcinoma (ASCC), and association with overall survival (OS) in univariate and multivariate analyses.

Heading	Patients (%)	Univariable Analysis		Multivariable Analysis		
	Patients (%)	Events (%) ^a^	OS (*p*-Value) ^a, b^	HR ^f^	95% CI ^h^	*p*-Value ^i^
*Total*	93 (100)	37 (40.9)				
***Age (years)*** <65 ≥65	60 (64.5) 33 (35.5)	24 (40.0) 13 (39.4)	0.32 (NS)			
***Gender*** Male Female	18 (19.4) 75 (80.6)	6 (33.3) 31 (41.3)	0.83 (NS)			
***Tumor differentiation*** Well/Moderate Poor	83 (89.2) 10 (10.8)	31 (37.3) 6 (60.0)	**0.014**	0.40 1	0.14–1.16	0.092 (NS)
***Basaloid contingent*** Present Absent	9 (9.7) 84 (90.3)	2 (22.2) 35 (41.7)	0.23 (NS)			
***HPV status*** Genotype 16 Other genotypes	85 (91.4) 8 (8.6)	33 (38.8) 4 (50.0)	**0.005**	1 6.78	1.66–27.8	**0.008**
***Concomitant HIV infection ^c^*** Yes No	8 (8.8) 83 (91.2)	5 (62.5) 31 (37.3)	0.099 (NS)	1.87 1	0.57–6.16	0.31 (NS)
***Tumor stage (AJCC 2010) ^a^*** I II IIIA IIIB IV	8 (8.9) 37 (41.1) 21 (23.3) 20 (22.2) 4 (4.4)	3 (37.5) 16 (43.2) 4 (19.0) 9 (45.0) 2 (50.0)	0.088 (NS)	1 2.49 2.82 8.04 9.07	0.49–12.7 0.18–43.5 0.44–146 0.55–148	**0.048**
***Lymph node invasion ^c^*** Yes No	40 (44.0) 51 (56.0)	15 (37.5) 20 (39.2)	0.094 (NS)	0.51 1	0.15–1.80	0.30 (NS)
***Synchronous metastasis ^d^*** Yes No	4 (4.3) 88 (95.7)	2 (50.0) 34 (38.6)	0.44 (NS)			
***Sample status*** Treatment-naive tumor Tumor recurrence	54 (58.1) 39 (41.9)	16 (29.6) 21 (53.8)	0.45 (NS)			
***Complete response after RT/CRT ^e^*** Yes No	72 (86.7) 11 (13.3)	25 (34.7) 7 (63.6)	**<0.0001**	0.16 1	0.00–0.46	**0.001**
***PIK3CA mutational status*** Wild-type Mutated	71 (76.3) 22 (23.7)	27 (38.0) 10 (45.5)	0.57 (NS)			

^a^ Data available for 90 patients, ^b^ Log Rank Test, ^c^ Data available for 91 patients, ^d^ Data available for 92 patients, ^e^ Data available for 83 patients, ^f^ Hazard Ratio, ^h^ 95% Confidential Interval, ^i^ Multivariate COX analysis, CRT: chemo-radiotherapy; RT: radiotherapy; NS: not significant, Significant results are displayed in bold; Factors with *p* < 0.10 in univariate analysis were selected for multivariate analysis.

**Table 2 cancers-11-01846-t002:** Association between mechanisms of integration of HPV and clinical, biological, and pathological characteristics of the 93 patients with HPV-positive anal squamous cell carcinoma (ASCC).

	Patients (%)	Number of Patients (%)			*p*-Value ^a^
Heading		*EPI*	*2J*	*MJ*	
*Total*	93 (100)	42 (45.2)	15 (16.1)	36 (38.7)	
***Age (years)*** <65 ≥65	60 (64.5) 33 (35.5)	28 (66.7) 14 (33.3)	10 (66.7) 5 (33.3)	22 (61.1) 14 (38.9)	0.86 (NS)
***Gender*** Male Female	18 (19.4) 75 (80.6)	7 (16.7) 35 (83.3)	3 (20.0) 12 (80.0)	8 (22.2) 28 (77.8)	0.83 (NS)
***Tumor differentiation*** Well/Moderate Poor	83 (89.2) 10 (10.8)	37 (88.1) 5 (11.9)	14 (93.3) 1 (6.7)	32 (88.9) 4 (11.1)	0.85 (NS)
***Basaloid contingent*** Present Absent	9 (9.7) 84 (90.3)	6 (14.3) 36 (85.7)	0 (0.0) 15 (100.0)	3 (8.3) 33 (91.7)	0.26 (NS)
***HPV status*** Genotype 16 Other genotypes	85 (91.4) 8 (8.6)	36 (85.7) 6 (14.3)	14 (93.3) 1 (6.7)	35 (97.2) 1 (2.8)	0.44 (NS)
***Concomitant HIV infection ^b^*** Yes No	8 (8.8) 83 (91.2)	3 (7.3) 38 (92.7)	2 (13.3) 13 (86.7)	3 (8.6) 32 (91.4)	0.92 (NS)
***Tumor stage (AJCC 2010) ^c^*** I II IIIA IIIB IV	8 (8.9) 37 (41.1) 21 (23.3) 20 (22.2) 4 (4.4)	5 (11.9) 18 (42.9) 9 (21.4) 8 (19.0) 2 (4.8)	1 (6.7) 6 (40.0) 3 (20.0) 4 (26.7) 1 (6.7)	2 (5.6) 13 (36.1) 9 (25.0) 8 (22.2) 1 (2.8)	0.98 (NS)
***Lymph node invasion ^b^*** Yes No	40 (44.0) 51 (56.0)	14 (33.3) 28 (66.7)	8 (53.3) 7 (46.7)	18 (52.9) 16 (47.1)	0.17 (NS)
***Synchronous metastasis ^d^*** Yes No	4 (4.3) 88 (95.7)	2 (4.8) 40 (95.2)	1 (6.7) 14 (93.3)	1 (2.9) 34 (97.1)	0.82 (NS)
***Sample status*** Treatment-naive tumor Tumor recurrence	54 (58.1) 39 (41.9)	25 (59.5) 17 (40.5)	9 (60.0) 6 (40.0)	20 (55.6) 16 (44.4)	0.93 (NS)
***Initial therapy*** *Upfront surgery ^d^* No Yes *RT or CRT ^b^* RT CRT With concomitant 5FU-MMC With concomitant 5FU-CDDP With other concomitant CT No RT/CRT	77 (83.7) 15 (16.3) 22 (24.2) 61 (67.0) 11 (18.0) 44 (77.0) 3 (4.9) 8 (8.8)	36 (85.7) 6 (14.3) 10 (24.4) 27 (65.9) 5 (18.5) 21 (77.8) 1 (3.7) 4 (9.8)	12 (80.0) 3 (20.0) 2 (13.3) 11 (73.3) 1 (9.1) 10 (90.9) 0 (0.0) 2 (13.3)	29 (82.9) 6 (17.1) 10 (27.8) 23 (63.9) 5 (21.7) 16 (69.6) 2 (8.7) 2 (5.6)	0.86 (NS) 0.76 (NS)
***Complete Response after RT/CRT ^e^*** Yes No	72 (86.7) 11 (13.3)	34 (91.9) 3 (7.1)	10 (76.9) 3 (23.1)	28 (84.8) 5 (15.2)	0.36 (NS)
***PIK3CA mutational status*** Wild-type Mutated	71 (76.3) 22 (23.7)	38 (90.5) 4 (9.5)	8 (53.3) 7 (46.7)	25 (69.4) 11 (30.6)	**0.0069**
***HPV viral load*** Low (<12) High (≥12)	41 (44.1) 52 (55.9)	21 (50.0) 21 (50.0)	10 (66.7) 5 (33.3)	10 (27.8) 26 (72.2)	**0.022**

^a^ Chi-Square Test; *p* values for comparison of the EPI group vs. the 2J group vs. the MJ group for each parameter; ^b^ Data available for 91 patients; ^c^ Data available for 90 patients; ^d^ Data available for 92 patients; ^e^ Data available for 83 patients; CDDP: cisplatin; CRT: chemo-radiation; CT: chemotherapy; MMC: mitomycin C; RT: radiation therapy. NS: not significant, Significant results are displayed in bold.

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
