# Peer review of "Mechanistic Signatures of Human Papillomavirus Insertions in Anal Squamous Cell Carcinomas"

_cancers, 2019, doi:10.3390/cancers11121846_

Round 1

Reviewer 1 Report

Morel et al report the characterization of a cohort of HPV+ anal squamous carcinoma patients.  They determine HPV type, integration status, viral load and correlate these with clinical and patient data.  

Overall, this looks like a solid cohort in a cancer type that is relatively understudied and has the potential to be an important additon to this area.  However, the manuscript suffers from a few flaws that collectively detract from this potential.  The most important of these are the lack of an appropriate multivariate analysis and a weak discussion.

1) Importantly, the authors need to perform a multivariate analysis to determine independent predictors of outcome. As one example, the lack of CR is associated with poorer outcomes (as expected, see Table 1) and this could just be skewing some of the comparisons of other clinical variables.  I would also like to see hazard ratios (with 95% confidence intervals) as well for the univariate and multivariate analyses.

2) The discussion just needs to be far more thorough. They are doing a disservice to themselves with the minimal discussion as presented. The authors need to work through each of there significant results in a logical matter, putting them into context with current data and pointing out novelty. For example, viral load predicts outcome (Table 2).  Tell me why.  At least speculate something logical (immune response to viral antigens?) and tell us if this has this been observed in other HPV+ cancers.

The manuscript also suffers from a large number of minor issues that and can be corrected with a little care and attention.

1) The figure legends are too minimal and uninformative.  They need to be expanded sufficiently that the reader can understand what the figure is showing without undue difficulty and reference to the main manuscript.

2) The methods are minimal.  The methods for all procedures should be described in at least a brief outline, rather than just a reference.

3) The results and discussion are in general too brief and lack any sort of context. For example, HPV16 is predictive of survival vs other HPV types from Table 1. The authers don't even mention this in the results.  This exact same result has been reported for both HPV+HNSCC and cervical cancer, and it is certainly interesting and noteworthy that the same thing happens in anal cancers.  This comparison between HPV dependent cancers located at different anatomical sites should also be brought forward in the discussion.

4) The quality of the figures seems low.  They need to be prepared at higher resolution.

5) There are some inconsistencies in the text like the use of "human papillomavirus" vs "human papilloma virus", the use of numbers as numbers (3) vs as text (three).  The authors should go over the manuscript carefully.

6) The referencing is minimal.  The authors should dig a little deeper for referencing in their field and related fields, particularly in the context of the intro and discussion.  For example, on page 7, they reference Olthof et al (2014), when other papers from the same time frame show the same thing, perhaps even better, like PMID:25313082.

7) It is not clear what comparisons the p values in Table 2 actually refer to.  The author likely require an FDR analysis given that multiple hypotheses are tested.

8) It looks like their are 3 integration events in chromosome 19 adjacent to the NFIX gene discussed.  This is probably worth discussing.

9) Some comparisons are probably underpowered. The authors might consider a post-hoc analyses (like G*power) to determine if their is no correlation simply because some of their comparison are underpowered.

10) The HIV patients may screw things up.  Are the clinical correlations the same or different if you exclude them?

11) If you exclude non-CRs from the analysis, are the clinical correlations the same or different?

Author Response

We would like to thank you and the Reviewers for their relevant and constructive comments and suggestions. We attach a revised version of the manuscript with tracked-in-red changes in the body of the text. Please find below the point-by-point responses to the Reviewers’ comments.

REVIEWER #1

1) Importantly, the authors need to perform a multivariate analysis to determine independent predictors of outcome. As one example, the lack of CR is associated with poorer outcomes (as expected, see Table 1) and this could just be skewing some of the comparisons of other clinical variables.  I would also like to see hazard ratios (with 95% confidence intervals) as well for the univariate and multivariate analyses.

We thank the reviewer for this valuable suggestion. We performed a multivariate analysis and added the risk ratios with 95% confidence intervals for each univariate and multivariate analysis factor (Cox model) in the revised version of Table 1 and in New Suppl. Table 3.

2) The discussion just needs to be far more thorough. They are doing a disservice to themselves with the minimal discussion as presented. The authors need to work through each of there significant results in a logical matter, putting them into context with current data and pointing out novelty. For example, viral load predicts outcome (Table 2).  Tell me why.  At least speculate something logical (immune response to viral antigens?) and tell us if this has this been observed in other HPV+ cancers.

We expanded the discussion to include comments on literature data for each primary finding, namely (1) viral load, (2) favorable prognostic value of HPV16, and (3) NFIX.

The manuscript also suffers from a large number of minor issues that and can be corrected with a little care and attention.

1) The figure legends are too minimal and uninformative.  They need to be expanded sufficiently that the reader can understand what the figure is showing without undue difficulty and reference to the main manuscript.

Thank you for this comment. We expanded the Figure legends according to reviewer’s suggestion.

2) The methods are minimal.  The methods for all procedures should be described in at least a brief outline, rather than just a reference.

Thank you for this comment. We expanded the description of Methods as suggested by the reviewer, in particular for DNA extraction, HPV genotyping and statistical analysis.

3) The results and discussion are in general too brief and lack any sort of context. For example, HPV16 is predictive of survival vs other HPV types from Table 1. The authers don't even mention this in the results.  This exact same result has been reported for both HPV+HNSCC and cervical cancer, and it is certainly interesting and noteworthy that the same thing happens in anal cancers.  This comparison between HPV dependent cancers located at different anatomical sites should also be brought forward in the discussion.

Thank you. As stated in response to main question 2, a discussion on the prognostic value of HPV16 has been added.

4) The quality of the figures seems low.  They need to be prepared at higher resolution.

The quality of Figures has been improved, they have been reformatted and uploaded in higher resolution.

5) There are some inconsistencies in the text like the use of "human papillomavirus" vs "human papilloma virus", the use of numbers as numbers (3) vs as text (three).  The authors should go over the manuscript carefully.

We apologize for these inconsistencies, the manuscript has been edited and the errors were carefully corrected throughout the text.

6) The referencing is minimal.  The authors should dig a little deeper for referencing in their field and related fields, particularly in the context of the intro and discussion.  For example, on page 7, they reference Olthof et al (2014), when other papers from the same time frame show the same thing, perhaps even better, like PMID:25313082.

We added 9 references including the article suggested by the reviewer.

7) It is not clear what comparisons the p values in Table 2 actually refer to.  The author likely require an FDR analysis given that multiple hypotheses are tested.

The p values ​​apply to the Chi-square test comparing the EPI group vs. the 2J group vs. the MJ group. This has been clarified in the caption below Table 2 as well as in Supplementary Table 2.

8) It looks like their are 3 integration events in chromosome 19 adjacent to the NFIX gene discussed.  This is probably worth discussing.

We thank the reviewer for raising this point. We indeed found 4 intragenic integration events in the NFIX gene as well as 3 other integration events adjacent to the NFIX gene but non-intragenic, of unknown significance. We modified Figure 2 and its legend to clarify this and inserted a sentence to comment on this in the Results section.

9) Some comparisons are probably underpowered. The authors might consider a post-hoc analyses (like G*power) to determine if their is no correlation simply because some of their comparison are underpowered.

Thank you for this comment. We used appropriate test to take into account the sample size, i.e. Chi-square test with Yates’ correction or Fisher’s exact test.

10) The HIV patients may screw things up.  Are the clinical correlations the same or different if you exclude them?

Thank you for the comment. We found the same significant clinical correlations in the non-HIV patient population (Results section and Suppl. Figure 3). This result has been added in the text.

11) If you exclude non-CRs from the analysis, are the clinical correlations the same or different?

Similar to the non-HIV population, we conducted additional specific analysis in CR and non-CR populations. The significant clinical correlations were the same (Results section and Suppl. Figure 3). This result has been added in the text.

Reviewer 2 Report

This manuscript evaluated the HPV signatures in patients with ASCC, and the association of HPV signature with OS. However, there’s concerns about the MS. Also, the manuscript needs to be further edited to make it fluent and logistic.

In the section of Patient characteristics, the description is not clear. How many patients were included in total? And what’s the different stratification? In Figure4-a, why the survival curve of NFIX insertion group ends in about Month 160, however the curve of no NFIX insertion group ends over month 240? Why did you use load 12 to distinguish Low HPV load and High HPV load? Why do you think HPV insertion into NFIX gene showed survival favorable effect on patients with ASCC? How about the PIK3C, c-myc, and P53 gene status of these patients? In Figure 3, did you subgroup the patients with ASCC via different treatment? In Figure 1, what’s the statistic method did you use to compare the HPV-positive anal cancer and cervical cancer? How about the HIV status of the patients with ASCC?

Author Response

REVIEWER #2

In the section of Patient characteristics, the description is not clear. How many patients were included in total?

Ninety-three patients were included, corresponding to 96 tumor samples (3 patients with 2 replicate samples). This is explained in the first paragraph of the Material and Methods section : « Ninety-six cryopreserved HPV-positive tumor biopsy specimens from 93 patients treated for ASCC at the Institut Curie Hospital between 1993 and 2017 were retrospectively retrieved from the institutional tissue bank. Matched initial diagnostic biopsy and recurrence biopsy samples were available for three patients.”

And what’s the different stratification?

We compared treatment-naive and recurrent disease groups for HPV integration : no difference was found (p=0.93). Following the comments of REVIEWER #1, we also added subgroup analyses (response to treatment, OS) according to treatment type (RT vs. CRT), CR vs. non-CR and HIV status (Results section and new Suppl. Figure 3).

In addition, for NFIX gene integration analysis were the analysis was carried out in the overall patient population and in the subgroup of patients with HPV integration (Figure 4).

In Figure4-a, why the survival curve of NFIX insertion group ends in about Month 160, however the curve of no NFIX insertion group ends over month 240?

We thank the reviewer for raising this point. The Figure 4 has been modified.

Why did you use load 12 to distinguish Low HPV load and High HPV load?

We added an explanation as to why the load 12 was selected as cut-off for HPV load (based on ROC curve, see « Methods » section).

Why do you think HPV insertion into NFIX gene showed survival favorable effect on patients with ASCC?

The Discussion on the favorable survival effect of HPV insertion into NFIX gene has been expanded (see response to REVIEWER #1).

How about the PIK3C, c-myc, and P53 gene status of these patients?

Mutations in the PIK3CA gene were studied because they had already been associated with HPV integration (Ref. #30); the results are presented in Suppl. Table 3. For c-Myc and P53, there was no available evidence of a major link between these immunohistochemical markers and HPV insertion and these analyzes were beyond the scope of this article.

In Figure 3, did you subgroup the patients with ASCC via different treatment?

Thank you for this suggestion. We subgrouped the patients according to treatment type (CRT vs. RT) and response (CR vs. non-CR) in the Results section and in an additional Figure (Suppl. Figure 3).

In Figure 1, what’s the statistic method did you use to compare the HPV-positive anal cancer and cervical cancer?

The p values ​​apply to the global Chi-square test comparing the different groups. This has been clarified in the caption below Figure 1.

How about the HIV status of the patients with ASCC? 

Thank you for the comment. See reply to REVIEWER #1, question 10. We found the same correlation in the HIV-negative patient population (Results section Suppl. Figure 3). This result has been added.

Round 2

Reviewer 1 Report

The revised manuscript is considerably improved and should have a stronger impact. I commend the authors for their efforts and congratulate them on a very nice study.

At the authors discretion, I would still recommend that they incorporate references related to HPV type having an impact on survival in other HPV dependent cancers. For HNSCC, PMID 27010835 and 27688111 would be appropriate.